# Visualization and Interpretation of Multivariate Associations with Disease Risk Markers and Disease Risk—The Triplot

**DOI:** 10.3390/metabo9070133

**Published:** 2019-07-06

**Authors:** Tessa Schillemans, Lin Shi, Xin Liu, Agneta Åkesson, Rikard Landberg, Carl Brunius

**Affiliations:** 1Cardiovascular and Nutritional Epidemiology, Institute of Environmental Medicine, Karolinska Institutet, SE-171 77 Stockholm, Sweden; 2Department of Biology and Biological Engineering, Chalmers University of Technology, SE-412 96 Gothenburg, Sweden; 3Department of Epidemiology and Biostatistics, School of Public Health, Xi’an Jiaotong University Health Science Center, SE-710049 Xi’an, China; 4Department of Public Health and Clinical Medicine, Umeå University, SE-901 87 Umeå, Sweden

**Keywords:** triplot, metabolomics, multivariate risk modeling, environmental factors, disease risk

## Abstract

Metabolomics has emerged as a promising technique to understand relationships between environmental factors and health status. Through comprehensive profiling of small molecules in biological samples, metabolomics generates high-dimensional data objectively, reflecting exposures, endogenous responses, and health effects, thereby providing further insights into exposure-disease associations. However, the multivariate nature of metabolomics data contributes to high complexity in analysis and interpretation. Efficient visualization techniques of multivariate data that allow direct interpretation of combined exposures, metabolome, and disease risk, are currently lacking. We have therefore developed the ‘triplot’ tool, a novel algorithm that simultaneously integrates and displays metabolites through latent variable modeling (e.g., principal component analysis, partial least squares regression, or factor analysis), their correlations with exposures, and their associations with disease risk estimates or intermediate risk factors. This paper illustrates the framework of the ‘triplot’ using two synthetic datasets that explore associations between dietary intake, plasma metabolome, and incident type 2 diabetes or BMI, an intermediate risk factor for lifestyle-related diseases. Our results demonstrate advantages of triplot over conventional visualization methods in facilitating interpretation in multivariate risk modeling with high-dimensional data. Algorithms, synthetic data, and tutorials are open source and available in the R package ‘triplot’.

## 1. Introduction

Environmental factors, such as diet, smoking, and pollutants, are associated with risk of developing non-communicable diseases (NCDs), including obesity, type 2 diabetes (T2D), and cardiovascular disease [1], which together constitute the leading cause of morbidity, mortality, and high healthcare costs worldwide. The role of lifestyle factors in development and progression of NCDs has often been studied in prospective cohorts or case-controlled studies, where associations of specific exposures with health outcomes or intermediate risk markers of NCDs (e.g., blood pressure, lipid profiles, and body weight) are assessed. Several challenges exist in the research on exposure–health relationships, including the measurement of environmental factors and the lack of understanding of underlying molecular mechanisms that are affected by the exposures [2]. 

Metabolomics is the comprehensive assessment of metabolites in biological samples, which enables investigation of physiological and biological states at the molecular phenotype level, reflecting both exogenous and endogenous exposures. Thus, metabolomics could potentially advance the understanding of associations between exposures and health status [3,4,5]. For example, using metabolomics to identify metabolite biomarkers objectively reflecting dietary exposures could provide a complement to self-reported dietary assessments that are known to suffer from large systematic and random measurement errors [6]. Metabolomics can also be used to link exposures to outcomes [7,8] by detecting endogenous changes in response to exposures [3]. However, in addition to these advantages, application of metabolomics in epidemiologic research makes interpretation and visualization of the results more complex due to the high dimensionality of the data.

Both multivariate analysis (e.g., reduced rank/component-based techniques) and univariate analysis are routinely used in metabolomics studies to extract meaningful information from complex datasets and thus provide biological knowledge of the research question under investigation [9]. Univariate analyses allow both for simultaneous investigation of multiple study factors, time series data, as well as adjustment for potential covariates or confounders. In general, univariate methods also provide more straightforward interpretation of results compared to multivariate analyses, which on the other hand make use of all variables simultaneously and are well-equipped to deal with high collinearity, which is often a challenge in epidemiological studies [10]. However, they offer limited options to investigate several study factors simultaneously, i.e., analyze data from time series or adjust for potential covariates or confounders. 

Results from metabolomics studies aiming to investigate exposure–disease relationships are often using a combination of figures to illustrate the findings. Observation scores and metabolite loadings from latent variable (LV) modeling (e.g., principal component analysis (PCA), factor analysis (FA), or partial least squares (PLS)) can be shown, e.g., in a biplot (Figure 1a), to identify outliers, to visualize separation of individuals into subgroups, and to examine how individual metabolites contribute to the LVs. Correlations between individual metabolites or LV scores and exposures are then frequently visualized using heatmaps (Figure 1b). Finally, individual metabolites or LV scores can be used as independent variables to model disease outcome or intermediate risk markers. Associated risk can then be visualized as odds ratios (ORs) or beta coefficients from logistic or linear regressions in a forest plot (Figure 1c). However, the lack of effective tools for direct interpretation of the relationship between exposures, metabolome, and outcome measure by visualization of combined data makes interpretation and communication of findings difficult. 

We therefore developed the novel ‘triplot’ tool to facilitate visualization and interpretation of multivariate risk modeling, which enables a global, combined overview of information representing the metabolome (or other types of multivariate data), exposures, or environmental factors of interest and associated health outcomes (i.e., disease outcomes or intermediate risk factors) (Figure 1d). We present the workflow of the triplot package and demonstrate its applicability using two synthetic datasets that were simulated from a case-controlled study nested within the Swedish prospective Västerbotten Intervention Programme cohort [11] and from a cross-sectional study of Carbohydrate Alternatives and Metabolic Phenotypes in Chinese young adults [12]. 

## 2. Materials and Methods

### 2.1. Synthetic Data 

*‘HealthyNordicDiet’*: This synthetic dataset was simulated from data used in a case-controlled study nested within the Swedish prospective Västerbotten Intervention Programme (VIP) cohort [8]. The entire study protocol was approved by the Regional Ethics Committee in Uppsala, Sweden (registration number 2014/011). The original study material was used to investigate how the plasma metabolome and the risk of developing T2D were related to compliance to a Healthy Nordic Diet [8]. Detailed information on study design and metabolomics data acquisition is provided elsewhere [7,8]. In brief, the original dataset included 421 participants from VIP at baseline (median time of 7 years before T2D diagnosis). Each case was individually matched to one nondiabetic participant on age, gender, sampling date, and sample storage time. Untargeted liquid chromatography quadrupole time-of-flight mass spectrometry (LC-qTOF-MS) metabolomics was performed on plasma samples using reverse phase and hydrophilic interaction chromatography in both positive and negative electrospray ionization modes. In total, 31 plasma metabolites related to a priori-defined healthy Nordic dietary indices, i.e., the Baltic Sea Diet Score (BSDS) and Healthy Nordic Food Index (HNFI), were selected using a random forest algorithm incorporated into a repeated double cross-validation framework with unbiased variable selection [8,13]. Subsequently, associations were investigated between the 31 dietary index-related metabolites, dietary intakes, and T2D risk [8]. 

The simulated data contains three data frames: Baseline characteristics of participants (*BaselineData*, 11 variables), identified metabolites associated with healthy Nordic diet (*MetaboliteData*, 31 variables), and food items associated with Healthy Nordic Diet (*FoodData*, 17 variables). The data frames are row-wise matched by observation and consist of 1000 synthetic observations that correspond to 500 case-controlled pairs matched by gender and age. 

*CAMP*: This synthetic dataset was simulated from real data used in a cross-sectional study of carbohydrate alternatives and metabolic phenotypes [12]. The study was approved by the ethical committee of Xi’an Jiaotong University Health Science Center, and all participants provided written informed consent. The original data were obtained from fasting plasma samples from 86 men and women that were between 18–35 years of age. Samples were analyzed by untargeted LC-qTOF-MS metabolomics using reverse phase chromatography in both positive and negative electrospray ionization modes. Associations were investigated between an optimal selection of plasma metabolites predictive of BMI and dietary intakes as well as several clinical measurements. 

The simulated data contains three data frames: Clinical measurements (*ClinData*, 11 variables), plasma metabolites predictive of BMI (*MetaboliteData*, 20 variables), and dietary intake as measured by food frequency questionnaires (*FoodData*, 11 variables). The data frames are row-wise matched by observation and consist of 300 synthetic observations.

### 2.2. Algorithm Description 

The ‘triplot’ is a novel tool that simultaneously integrates three levels of information, effectively providing interpretable visualization of multivariate associations between exposures, metabolome, and disease risk by superimposing LVs from multivariate modeling of, e.g., metabolomics data with correlations of exposures (or other correlations) and associations with disease risk or intermediate risk markers (Figure 2). An overview of the functions and workflow of the triplot package is presented in Table 1. 

In the first layer of the triplot, LV modeling is performed on a high-dimensional dataset, generated from, e.g., metabolomics, proteomics or other omics, to reduce the data dimensionality, and to aggregate correlated variables into LVs. The choice of LV modeling method depends on the preference of the user, the data, and the analytical question. The triplot algorithm accepts input from any LV modeling that conforms to reporting observation scores and variable loadings. Frequently used LV algorithms include unsupervised PCA and FA as well as supervised PLS analyses. PCA and FA are used to describe the total variability among the observed (metabolomics) variables using a lower number of LVs called principal components or factors, respectively. PLS is conceptually similar but identifies components that are instead optimized for covariation between the observed (independent) variables and an outcome (dependent) variable [14,15].

There are several methods to determine the number of LVs to retain in unsupervised LV modeling, i.e., PCA and FA [16]. Among them, a scree plot, which shows how much variation each factor or principal component captures from the data, and very simple structure analysis are commonly used [17]. For supervised modeling like PLS, the number of LVs should be optimized by cross validation. Out of several cross-validation approaches, repeated double cross validation has shown advantages in estimation of the optimum number of PLS components and estimations of prediction errors over several other commonly used validation approaches, such as k-fold and leave-n-out [13,18,19].

A second layer presents correlations between LV observation scores and single or multiple exposures, such as self-reported dietary intakes. Correlation coefficients can be obtained by any correlation methodology that is suitable for the data structure, such as the Pearson method for linear correlations, the Spearman method for non-linear (rank) correlations, or polychoric/polyserial correlation methods for ordinal variables [20]. In order to adjust the correlation results for confounders, users can also apply partial correlations [21]. 

Associations between the LV observation scores and disease risk or intermediate risk factors are added in a third layer. Users can define risk associations suitable for different study designs, such as ORs of disease risk calculated using (conditional) logistic regression in case-controlled studies, hazard ratios of disease risk calculated using cox regression in prospective cohorts, or beta coefficients of intermediate risk markers calculated using linear regression in cross-sectional studies. 

Associated correlations and risk estimates are added to the LV modeling in a modular, easy-to-use workflow, and a summarized overview in the form of a heatmap can then be generated to assist in selecting LVs to investigate using the triplot function.

### 2.3. Software and Implementation

The triplot algorithm is publicly available in an open source R implementation (https://gitlab.com/CarlBrunius/triplot). The repository provides the ‘triplot’ R package, installation instructions, synthetic data, and a tutorial that covers the case studies described in this manuscript in a high level of detail, as well as several additional case studies.

## 3. Results and Discussion

We applied various analyses on the two simulated datasets available from the package to demonstrate the wide applicability of the triplot. Disease risk (discrete outcome) is modeled using the ‘*HealthyNordicDiet*’ dataset and intermediate risk markers (continuous outcomes) are modeled using the ‘*CAMP*’ dataset.

### 3.1. HealthyNordicDiet

The original study was set up to explore plasma metabolites that could objectively reflect healthy Nordic dietary patterns in a matched case-controlled study and to assess associations between such patterns and later development of T2D [8]. The processing pipeline for the generation and visualization of the original data is described in Appendix A. A global overview of intercorrelations between plasma metabolites related to the healthy Nordic diet, dietary intake variables, as well as T2D risk is shown in Figure 3 (Tutorial—Example 1). PC1 constituted a metabolite profile, which directly reflected the healthy Nordic dietary indices and individual food components of the indices and was not associated with T2D risk after adjustment for lifestyle-related factors. PC2 instead, while it was negatively correlated with the healthy Nordic dietary indices, it was predominantly correlated with foods not part of the indices, e.g., margarine, sausages, and poultry, and also more strongly associated with risk of developing T2D, even after adjustment for BMI and lifestyle-related factors (smoking status, education, physical activity, and total energy intake). Results from different risk modeling approaches can easily be incorporated into the triplot framework, e.g., using normal logistic regression, which achieved similar OR estimates as conditional logistic regression (Figure 3, Tutorial—Example 2). 

Importantly, all information incorporated in the triplot visualization could have been obtained using conventional tools, such as separate PCA biplots, heatmaps, and forest plots for risk estimates (Figure 1). However, such an approach presents results scattered across different tables and/or figures, which impedes the direct interpretation of the results. The triplot algorithm instead provides an integrated overview of metabolites as well as associated exposures and risk estimates, which intuitively and clearly presents relevant biological information: The results obtained from synthetic data, i.e., that the metabolite profile related to healthy Nordic diet is not associated with T2D whereas that of more unhealthy dietary choices is, effectively mirror those that were obtained from authentic data [8].

### 3.2. CAMP

Obesity has been associated with increased morbidity and mortality from NCDs, and high BMI has also been associated with the intake of unhealthy food, i.e., fast food and red/processed meat [22,23]. The cross-sectional study of Carbohydrate Alternatives and Metabolic Phenotypes in Chinese young adults was therefore designed to assess relationships between diet, metabolic profiles, and risk factors of metabolic diseases, using both traditional epidemiological approaches and metabolomics techniques [12]. 

Intercorrelations between BMI-related plasma metabolites, dietary intakes, and metabolic traits are shown in Figure 4 (Tutorial—Example 3). The PCA-based triplot shows that the metabolite profile predicting BMI was strongly associated with liver enzyme activity, i.e., gamma-glutamyltransferase (GGT), alanine aminotransferase (ALT), and aspartate aminotransferase (AST) (Figure 4), and also with several other health-related metabolic traits, including fasting glucose, triglycerides, total cholesterol, as well as high- (HDL) and low-density lipoprotein (LDL) cholesterol (data not shown), in line with previous studies [24,25,26,27]. PC1 reflected metabolites that were positively associated with BMI and also correlated with a high intake of meat and refined grains and negatively with seafood intake, in agreement with observational studies [28,29,30]. We also found that a high intake of fruits correlated with BMI-related metabolites and other metabolic traits (Figure 4). Fruit consumption is widely considered an important part of a healthy diet, which may provide a host of beneficial nutrients, i.e., vitamins and minerals, dietary fiber, and polyphenols, and aid in the reduction of energy intake and body weight. However, conflicting results exist regarding associations between fruit intake and risk factors of NCDs, including BMI [31,32], as supported by the present investigation. Moreover, PC2 contained high negative loadings of, e.g., phosphatidylcholines containing the marine polyunsaturated fatty acid (C22:6), which in turn correlated positively with seafood intake. From the direct associations of PC2 with liver enzyme activity we then may speculate that the results support the benefits of seafood intake, rich in omega-3 polyunsaturated fatty acids, on human health [33,34,35,36]. 

Of note, the triplot can also be easily constructed based on components derived from supervised modeling of multivariate data (Tutorial—Example 4). To illustrate the wide applicability of the triplot, we performed PLS modeling on the BMI-related metabolites and assessed associations between PLS-derived metabolite components with dietary intakes and metabolic traits, which resulted in similar results as the PCA analysis (Appendix A). The overall direction of the associations obtained using synthetic data was comparable to the results that were obtained from authentic data, although the association between PC1 and high intake of meat was not significant in either synthetic or authentic data (Appendix A). 

## 4. Conclusions

In this work, we have proposed a novel tool, the ‘triplot’, which can be effectively used to visualize and interpret multivariate risk modeling with high-dimensional data. The framework for integration of metabolomics data, analyzed using either unsupervised or supervised LV modeling, with dietary intakes and disease risk or intermediate risk factors was illustrated using two synthetic datasets representing different study designs. Moreover, our results demonstrate how the triplot could provide advantages over conventional methods in terms of visualization and interpretation of modeling results and thus has the potential to assist in extracting biological meaning from complex data.

## Figures and Tables

**Figure 1 metabolites-09-00133-f001:**
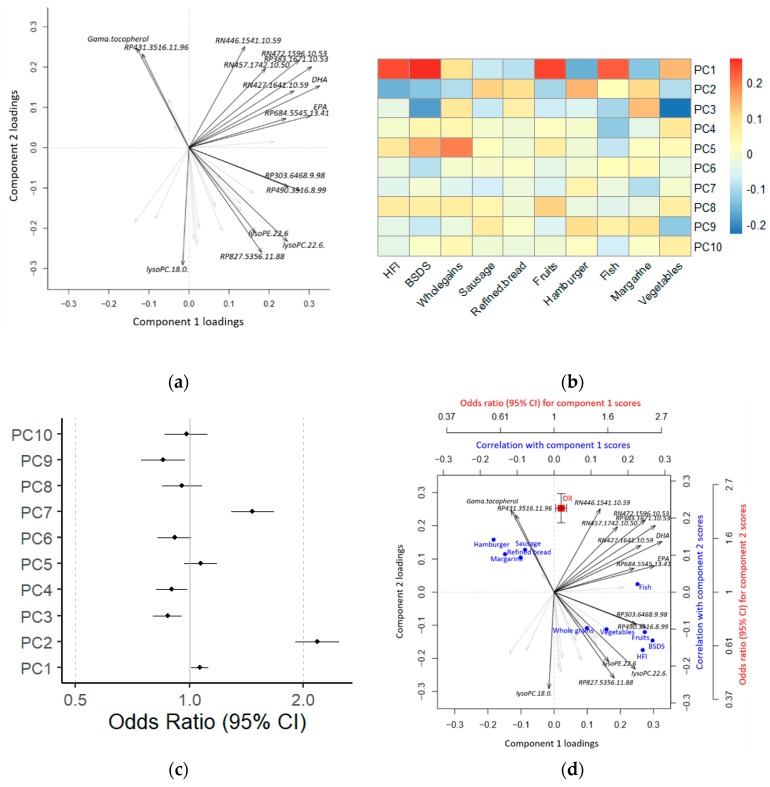
Using metabolomics data to investigate the relationship between exposures and disease risk. In a standard approach, latent variable (LV) modeling (**a**) is used to achieve a reduced rank approximation of the metabolomics data. Correlation heatmaps (**b**) and forest plots (**c**) are then used to associate observation scores with exposures and risks, respectively. For more direct interpretation, LV modeling, and their associations with exposures and risks can be visualized jointly in a triplot (**d**). HFI: Healthy Food Index; BSDS: Baltic Sea Diet Score; PC: Principal Component; OR: Odds Ratio; CI: Confidence Interval. lysoPC: Lysophosphatidylcholine; lysoPE: Lysophosphoethanolamine; EPA: Eicosapentaenoic acid; DHA: Docosahexaenoic acid; RP: reverse phase chromatography positive mode ionization; RN: reverse phase chromatography negative mode ionization.

**Figure 2 metabolites-09-00133-f002:**
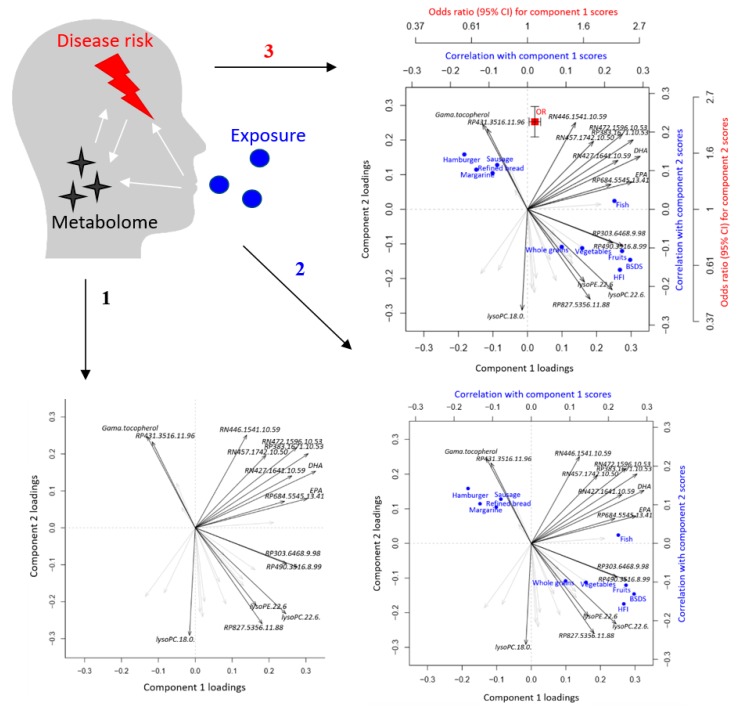
Link between exposures, metabolome, and disease risk as presented in the triplot. The first step consists of latent variable modeling of metabolomics data providing scores and loadings. The second step superimposes correlations between component scores and exposures (or covariates). The third step superimposes risk of outcome associated with the component scores.

**Figure 3 metabolites-09-00133-f003:**
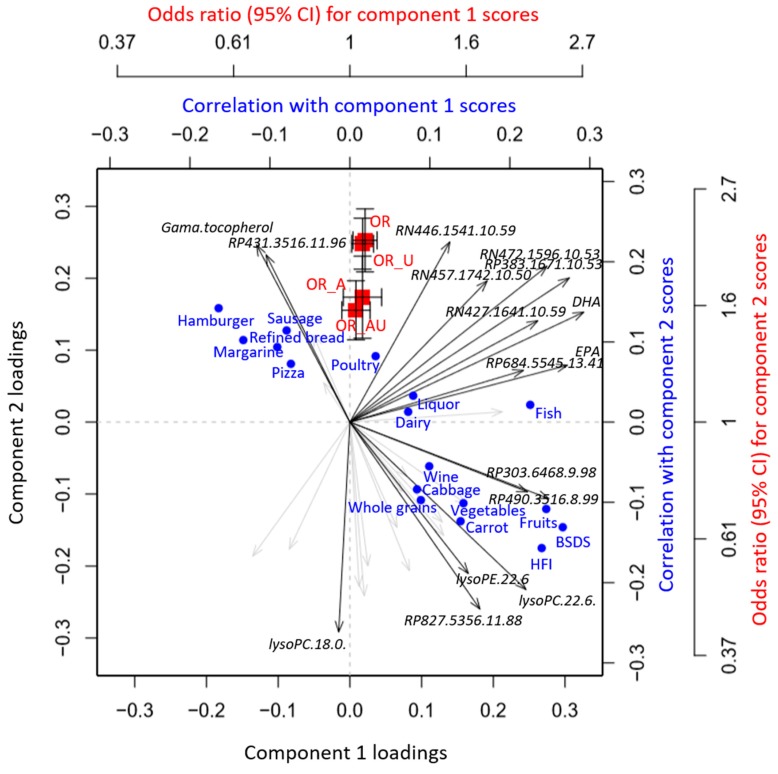
A PCA-based triplot visualizing the inter-correlation between plasma metabolites related to healthy Nordic diet and dietary intake variables as well as association with type 2 diabetes (T2D) risk. Odds ratios of T2D were calculated using conditional logistic regression with or without adjusting for BMI and lifestyle-related confounders (smoking status, education, physical activity, and total energy intake) (OR_A and OR, respectively). Risk associations were calculated similarly using unconditional logistic regression (OR_AU and OR_U, respectively). Correlations between PCA components and dietary intakes were calculated using partial Pearson method, adjusted for case-controlled status, gender, age, BMI, and lifestyle-related confounders (smoking status, education, and physical activity). Only metabolite feature loadings > 0.25 and dietary intake variables significantly correlated with the PCA components are shown.

**Figure 4 metabolites-09-00133-f004:**
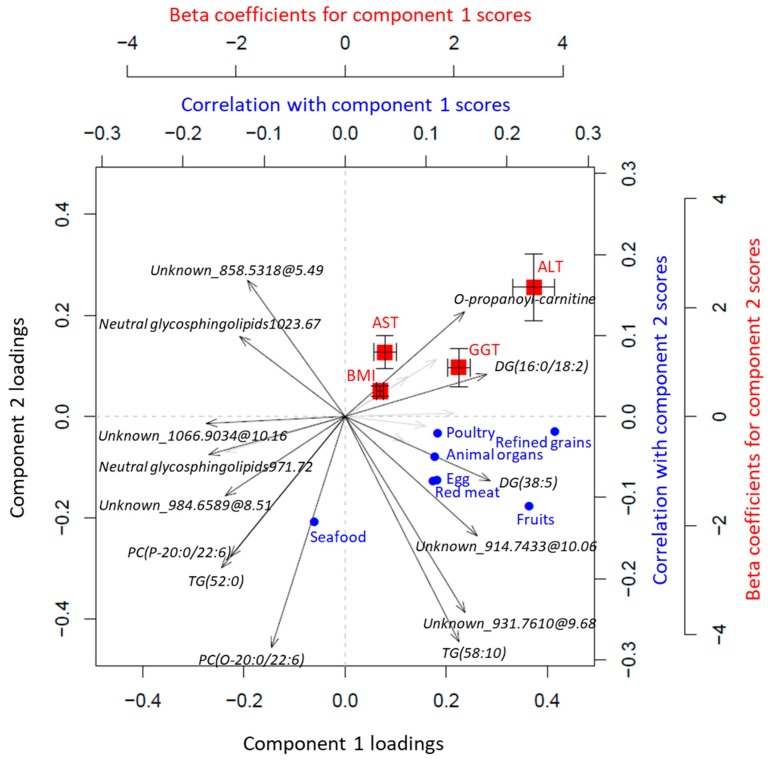
A PCA-based triplot visualizing the inter-correlations between plasma metabolites predicting BMI, dietary intake variables, and metabolic traits, after adjusting for age and gender. Correlations between PCA components and dietary intakes estimated from food frequency questionnaires were calculated using the partial Spearman method. Associations of PCA components with metabolic traits were assessed using linear regression. Only metabolite feature loadings > 0.25, significant correlations, as well as correlations with animal derived foods and metabolic traits with strongest associations are shown. ALT: alanine aminotransferase; AST, aspartate aminotransferase; GGT: gamma-glutamyltransferase.

**Table 1 metabolites-09-00133-t001:** Overview of the workflow and main functions in the ‘triplot’ R package.

Function	Description
First layer: Latent variable (LV) modeling	
*Custom ^a^*	Perform LV modeling of high-dimensional (metabolomics) data.
*makeTPO() ^a^*	Initiate a triplot object (TPO) from LV model
Second layer: Correlations	
*makeCorr()* *or custom ^b^*	Perform correlation analysis between LV observation scores and exposures or covariates.
*addCorr()*	Add correlation results to the TPO.
Third layer: Associated risk	
*crudeCLR(),* *crudeLR(),* *or custom ^c^*	Calculate risk associations (i.e., odds ratio or hazard ratio) in (conditional) logistic regression or association with intermediate risk markers (i.e., beta coefficient) in linear regression.
*addRisk()*	Add risk associations to the TPO.
Visualizations	
*checkTPO()*	Generate a heatmap visualizing correlations and risk associations to identify relevant LVs for the triplot visualization.
*triplot()*	Create a triplot containing LV analysis results, correlations, and risk associations.

^a^ Actual LV modeling is purposely omitted from the triplot package to give the user the choice of LV method, such as PCA, FA, or PLS. The *makeTPO()* function will accept any input that conforms to scores and loadings. ^b^
*makeCorr()* constitutes a convenience function for standard correlation analysis (Pearson, Spearman, Kendall). Partial correlation requires custom scripts and is covered in the tutorial. ^c^
*crudeLR()* and *crudeCLR()* constitute convenience functions for (conditional) logistic regression. Adjusting associations for covariates requires custom scripts and is covered in the tutorial.

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
