# Peer review of "Visualization and Interpretation of Multivariate Associations with Disease Risk Markers and Disease Risk—The Triplot"

_metabolites, 2019, doi:10.3390/metabo9070133_

Round 1
Reviewer 1 Report
The authors describe a novel visualization tool for metabolomics, which they call ‘triplot that can be downloaded and applied as R package. With this tool, they aim to overcome the separation of observational figures like e.g. score and loadings plots or heatmaps and interpreting figures where e.g. disease outcome is shown in dependence of individual metabolites. Two synthetic datasets are used as example to demonstrate their approach. This tool may help streamlining metabolomics reports, thus I recommend to publish the manuscript after some revision of the text.
As at least one of the original publication is not open access, it was cumbersome to keep track of the performed analysis that serves as example to demonstrate the suitability of their tool. I tried to download the program but the tutorial was not yet applicable. I would appreciate if - to better understand the steps of the program - the authors sketched in detail the data structure that they used and what analysis they performed. In particular in the case of “Healthy Nordic Diet”: which data were present in their data sets – is “food not part of the indices” part of the baseline characteristics? Are these data not used for PCA analysis as I presume? Are they used for heatmaps? Or only for interpretation? Once this part becomes clearer one can better appreciate the suggested imprvements.
Author Response
We thank the reviewer for the valuable feedback. We have now corrected the tutorial for the ‘triplot’ package and ensured its online availability: The “readme.md” (available from the repository page) has a working link to a tutorial in markdown format which we have confirmed renders well on the repository. We have furthermore confirmed that the tutorial renders in html format upon installation with the new instructions, i.e. devtools::install_gitlab("CarlBrunius/triplot", build_vignettes=T). Within the tutorial we also sketch in great detail the steps of the program, the data structure used and the analysis performed.
Furthermore, we have now included an additional figure in the supplementary data that clarifies the data structure and study design of the original study. The “foods not part of the indices” were measured at baseline and are part of the “FoodData” data frame. These food variables were not a part of the Healthy Nordic Food Index or Baltic Sea Diet Score, but they are nonetheless among the food items significantly associated with the Healthy Nordic Diet metabolome. A key point in the original paper is, in fact, that single indices may not accurately capture the complexity of dietary exposures and that other dietary choices will influence metabolomic profiles originally selected as potential biomarker candidates. In this particular instance, it seems that the metabolome of healthy diet also reflects poorer lifestyle choices, which in turn are more strongly associated with disease risk.
For the PCA analysis, only identified metabolites associated with the indices (available in the MetaboliteData data frame) are used. Baseline characteristics or food items are not used for the PCA analysis. Instead, baseline characteristics and food items are partial-correlated with the PC scores in the 2nd layer of the triplot. Baseline characteristics are again used for adjustment in logistic regression models to generate risk scores for type 2 diabetes. A heatmap is created to visualize the correlations and risk scores of several principal components in order to help the user to choose the most relevant principal components to visualize in the triplot tool.
Reviewer 2 Report
The manuscript "Visualization and interpretation of multivariate associations with disease risk markers and disease risk – The triplot" by Schillemans et al., is very well written and presents an interesting tool to assist an integrative overview of multivariate data. This is specially useful in the light of current high throughput "omics" data, combined with additional observations of high-dimensional data. Triplot provides the framework to assess the correlations in a simplified pipeline, that can be adapted to different types of data, as validated in the examples provided.
I believe the tool will be useful to the broad readership of "Metabolites", and other fields dealing with integrative high-dimensional data analysis. I support the acceptance for publication in its current form.
Author Response
We thank the reviewer for the valuable, positive feedback. We are glad to hear that you find this tool, which has been very useful to us in visualizing complex associations between metabolome, exposures and risk, helpful.